# Impact of Radiotherapy on Endotracheal Intubation Quality Metrics in Patients with Esophageal Cancer: A Challenge for Advanced Airway Management?

**DOI:** 10.3390/cancers16142540

**Published:** 2024-07-15

**Authors:** Davut D. Uzun, Timo Tryjanowski, Nathalie Arians, Stefan Mohr, Felix C. F. Schmitt, Christoph W. Michalski, Markus A. Weigand, Juergen Debus, Kristin Lang

**Affiliations:** 1Department of Anesthesiology, Medical Faculty Heidelberg, University Heidelberg, 69120 Heidelberg, Germany; deniz.uzun@med.uni-heidelberg.de (D.D.U.); stefan.mohr@med.uni-heidelberg.de (S.M.); felix.schmitt@med.uni-heidelberg.de (F.C.F.S.); markus.weigand@med.uni-heidelberg.de (M.A.W.); 2Department of Radiation Oncology, Medical Faculty Heidelberg, University Heidelberg, 69120 Heidelberg, Germany; timo.tryjanowski@gmail.com (T.T.); nathalie.arians@med.uni-heidelberg.de (N.A.); juergen.debus@med.uni-heidelberg.de (J.D.); 3Heidelberg Institute of Radiation Oncology (HIRO), 69120 Heidelberg, Germany; 4National Center for Tumor Diseases (NCT), 69120 Heidelberg, Germany; 5Department of General, Visceral and Transplantation Surgery, Medical Faculty Heidelberg, University Heidelberg, 69120 Heidelberg, Germany; christoph.michalski@med.uni-heidelberg.de; 6Heidelberg Ion-Beam Therapy Center (HIT), Department of Radiation Oncology, Medical Faculty Heidelberg, University Heidelberg, 69120 Heidelberg, Germany; 7Clinical Cooperation Unit Radiation Oncology, German Cancer Research Center (DKFZ), 69120 Heidelberg, Germany; 8German Cancer Consortium (DKTK), Partner Site Heidelberg, 69120 Heidelberg, Germany

**Keywords:** esophageal cancer, radiotherapy, radiation toxicity, advanced airway management, laryngoscopy, tracheal intubation, videolaryngoscopy

## Abstract

**Simple Summary:**

Radiotherapy is an important treatment option for esophageal cancer, in addition to surgery and chemotherapy. Ionizing radiation can cause alterations in the patient’s anatomy, particularly in the larynx area, which may make advanced airway management more challenging for anesthetists. The existing literature contains data on other entities of cancer and radiotherapy and their impact on advanced airway management. However, there is a lack of data on esophageal cancer. We retrospectively analyzed patients with esophageal cancer who underwent radiotherapy followed by surgery between 2012 and 2023 in our university hospital. It has been shown in the literature that post-radiotherapy effects can increase the risk of difficult endotracheal intubation and airway complications in anesthesiology. However, our study did not indicate any evidence of impaired advanced airway management in patients with esophageal cancer after radiotherapy.

**Abstract:**

(1) Background: Currently, no data are available in the literature investigating the influence of radiotherapy (RT) on endotracheal intubation success in patients with esophageal cancer. This study aims to evaluate the impact of RT on endotracheal intubation quality metrics in patients with esophageal cancer. (2) Methods: Patients with esophageal cancer who underwent RT followed by surgery between 2012 and 2023 at the University Hospital Heidelberg, Germany, were retrospectively analyzed. (3) Results: Fifty-five patients, predominantly males 65.5% with a mean age of 64 years, were enrolled. Overall, 81.8% of the patients had an ASA class of III, followed by 27.2% ASA II. The mean prescribed cumulative total dose to the primary tumor and lymph node metastasis was 48.2 Gy with a mean single dose of 1.8 Gy. The mean laryngeal total dose was 40.0 Gy. Direct laryngoscopy was performed in 80.0% of cases, followed by 12.1% videolaryngoscopy, and 7.2% required fiberoptic intubation. Overall, 96.4% of patients were successfully intubated on the first attempt. (4) Conclusions: It has been demonstrated that post-RT effects can increase the risk of airway management difficulties and complications. The results of our study did not indicate any evidence of impaired advanced airway management in patients with esophageal cancer who had undergone RT.

## 1. Introduction

The prevalence of cancer patients is steadily increasing, a trend that is also linked to demographic changes in the population [1,2]. Because of advances in treatment and early detection strategies, the number of people living with cancer and surviving in the long term is likely to increase in the future. As a result of this trend, it is more likely that cancer patients will require further surgeries in the future, necessitating advanced airway management [3].

A large population of patients with solid tumors require neoadjuvant therapy, typically involving radiotherapy (RT), prior to surgical tumor resection. Esophageal cancer is the sixth leading cause of cancer-related deaths, and its incidence is increasing worldwide [4,5]. Previously, squamous cell carcinoma (SCC) was the most prevalent form of cancer in the esophagus. However, in industrialized countries, the incidence of adenocarcinoma (AC) is increasing [6]. A change in risk factors is probably one reason for this increase [7]. More than two-thirds of patients are diagnosed with locally advanced or metastatic disease, and the 5-year overall survival rate is relatively low, ranging between 15% and 25% [8]. RT is an important treatment option for esophageal cancer, in addition to surgery and chemotherapy [9]. The treatment of esophageal cancer depends mainly on the clinical stage of the tumor and requires a multidisciplinary assessment. Neoadjuvant chemoradiotherapy improves overall survival in patients with resectable locally advanced esophageal cancer compared with surgery alone [10]. Modern radiation therapy options, such as intensity-modulated radiotherapy (IMRT), allow for more precise targeting of tumors, reducing radiation exposure to nearby organs and structures.

However, it is important to note that RT is also associated with side effects [11]. There has been limited research conducted on the airway changes induced after RT in patients with esophageal cancer. Ionizing radiation can cause changes in the tissue and thus alterations in the patient’s anatomy, particularly in the larynx area, which may make advanced airway management more challenging [11].

During the early phase, which usually lasts for several weeks, patients may experience dysphagia, dry mouth, mucositis, and dermatitis [12]. The difficulties are mainly caused by pathophysiological changes following RT. Additionally, edema caused by RT can make mask ventilation and advanced airway management challenging [13]. The altered anatomy due to fibrosis, restricted mouth opening, and trismus can severely impede conventional laryngoscopy, making airway securing extremely difficult. These effects are dose-dependent, and nowadays, it is potentially possible to reduce these side effects with modern RT techniques and thus better normal tissue sparing. Changes in the glottis, such as the formation of edema, can also make it difficult to distinguish among anatomical structures clearly [13]. Anesthetists involved in treating patients after RT should anticipate potential difficulties in airway management and prepare accordingly. A comprehensive preoperative airway assessment during the anesthetic consultation and careful planning of airway management are crucial to identify patients with difficult tracheal intubation (DTI). According to the updated 2022 ASA Guidelines for the management of difficult airways, a ‘failed or difficult tracheal intubation’ is defined as “a tracheal intubation requiring multiple attempts or a failed tracheal intubation after multiple attempts”. According to the ASA guidelines, the case was categorized as difficult intubation because of more than one intubation attempt [14]. In addition to the number of attempts, we also decided to consider the criteria. To broaden the classification of difficult intubation, cases with perioperative or delayed airway complications should be considered as an indicator of underlying or obvious difficulties with intubation. If the airway is potentially severely compromised, procedures such as awake fiberoptic endotracheal intubation should be considered early on [11].

In the current literature, no data are available about the effects or impact of RT on advanced airway management procedures in patients with esophageal cancer. Therefore, the aim of this study is to examine endotracheal intubation quality metrics related to post-radiotherapy among patients with esophageal cancer.

## 2. Materials and Methods

This retrospective study was performed following institutional guidelines and the Declaration of Helsinki of 1975 in its most recent version. Ethical approval for this study was given by the local ethics committee at University Hospital Heidelberg (S-123/2021).

### 2.1. Patient Selection

Patient selection was based on a retrospective database query of the Department of Radiation Oncology at the Heidelberg University Hospital. Patients aged older than eighteen who received radiation or chemoradiation therapy for esophageal cancer (adenocarcinoma or squamous cell carcinoma) of any stage with curative intention between 2012 and 2023 and following surgery with necessary endotracheal intubation were included in this analysis. This initial database query of all patients with inclusion criteria resulted in a total number of 224 patients. All patients with advanced airway management were included in the present study, regardless of whether the surgery took place years later or not. Exclusion criteria were esophageal cancer in the lower part, adenocarcinomas of the gastroesophageal junction, missing surgery and missing advanced airway management, history of malignancies within three years before therapy, pre-irradiation, death before surgery, or incomplete data. Patient confidentiality was maintained by anonymizing patient data to remove any identifying information. Thus, patient consent was not required. After the exclusion of the above criteria, 55 patients were included in the final analysis. Details are shown in Figure 1.

### 2.2. Treatment Characteristics

RT was based on CT-planned intensity-modulated radiotherapy (IMRT) as helical IMRT (TomoTherapy^®^, Accuray, Sunnyvale, CA, USA) or three-dimensional radiotherapy at the Heidelberg University Hospital. The Department of Anesthesiology was responsible for general anesthesia and advanced airway management for surgery. Prior to surgery, our anesthesia experts assessed all patients during the obligatory pre-anesthesia consultation and examination. This is where the patient’s complete medical history and anesthesia-related findings were screened and documented personally by our anesthesiologists.

The prescribed cumulative RT-dose to primary tumor and lymph node metastasis was 48.2 Gy (range: 41.4–58.8 Gy). The median single dose to primary tumor and lymph node metastasis was 1.8 Gy (range:1.8–3.0 Gy). The gross tumor volume (GTV) included all macroscopic tumors visible on the planning CT, including suspected nodal disease. The clinical target volume (CTV) was created by adding margins to the GTV (radial 0.5–1 cm, craniocaudal: 4–5 cm) and adjusted for the lymphatic drainage areas within this expansion. In cervical or proximal tumors, supraclavicular nodes were included. For planning target volume (PTV), a margin of 0.5–1 cm was added to create the planning target volume. In the present study, the larynx was subsequently contoured for all patients, a radiotherapy plan was drawn up, and the dose to the larynx was calculated precisely. Figure 2 demonstrates an example of a radiation plan for a 52-year-old patient with esophageal cancer in the upper esophagus. The larynx was marked in yellow as an organ at risk in this plan and the maximum dose was 51.04 Gy. Concomitantly, the chemotherapeutic regime included cisplatin/5-FU or FOLFOX in all patients.

### 2.3. Statistical Analysis

Data collection for the presented project was conducted using an electronic database system, specifically Microsoft Excel from Microsoft GmbH in Unterschleißheim/Germany. Detailed descriptive statistics are provided for all data collected. For continuous data and scores, the mean, standard deviation, minimum, median, and maximum were calculated. Statistical analyses for mean and standard deviation (SD) were performed using the statistical software IBM SPSS software version 24.0.

## 3. Results

The present study included patients with esophageal cancer who had undergone radiotherapy and subsequently required advanced airway management during subsequent surgeries. A total of fifty-five patients were included in the current study, who were predominantly male 65.5% with a mean age of 64 years (range: 49–82 years).

The mean Body Mass Index (BMI) was 23.6 kg/m^2^ (range: 15.3–36.3 kg/m^2^). Patient demographics and characteristics are shown in Table 1.

In addition, 96.3% of the patients had squamous cell carcinoma, and 3.7% of the patients had adenocarcinoma. The majority of the patients (69.1%) presented with esophageal cancer at stage T3. The detailed pathological characteristics are listed in Table 1. The tumor localization was in the mean at 21 cm (range 10–33 cm) tooth row. In the study cohort, 85.5% of the patients received treatment with IMRT, and 14.5% were treated with 3D-CRT. The mean prescribed cumulative total dose to the primary tumor and lymph node metastases was 48.2 Gy (range: 41.4–58.8 Gy), with a mean single dose of 1.8 Gy. The mean Dmax laryngeal total dose was 40.0 Gy (range 14.3–68.0 Gy). The mean PTV volume was 844.6 ccm (range 322–1322 ccm).

The mean time from RT to surgery was 334.7 days (range 20–3787 days). No evidence of esophago-tracheal fistulas was identified in the collective examined. RT treatment characteristics are shown in Table 2. Furthermore, the majority of patients exhibited relevant cardiovascular risk factors, predominantly hypertension (47.2%) and smoking (56.4%). The detailed distribution of cardiovascular risk factors is shown in Table 1.

Overall, 81.8% of the patients had an ASA class of III, followed by 27.2% ASA II, Table 3. Direct laryngoscopy was performed in 80.0% of the cases, followed by 12.1% video laryngoscopy, and 7.2% required fiberoptic endotracheal intubation. Overall, 78.2% of the patients had Cormack/Lehane (C/L) grade I, followed by 16.4% with C/L grade II and 5.4% with C/L grade III. In the current study, 96.4% of the patients were successfully intubated on the first attempt. The medications used for general anesthesia are listed in Table 3. Analysis of complications showed oropharyngeal bleeding in 1.8% of the patients analyzed, with no other documented complications such as severe hypoxemia, aspiration, or mortality, Table 4.

## 4. Discussion

To the best of our knowledge, this study represents the first evaluation of the impact of radiotherapy on endotracheal intubation quality metrics in patients with esophageal cancer. The fact that RT influences the patient’s anatomy is not new, and this can pose major challenges for the anesthetist in advanced airway management. Post-RT anatomical changes and potential difficulties with advanced airway management are not only challenging for the anesthetist but also pose a significant risk to the patient [15]. Despite the significant advancements in general anesthesia over recent decades, failures in advanced airway management remain the main cause of anesthesia-related cardiac arrest.

In a recent study from the U.K. by Cook et al., a cumulative 57.6% of perioperative cardiac arrests were due to airway failure and aspiration [15]. Some patient groups appear to be particularly affected by this. Ear, nose, and throat (ENT) surgery and abdominal surgery appear to have a higher risk of complications from advanced airway management [15]. The existing literature only contains data on the effects of RT and advanced airway management for head and neck tumors. For instance, patients undergoing RT for nasopharyngeal carcinoma (NPC) appear to have significantly higher rates of difficult tracheal intubation (DTI) and intubation failure than the general surgical population [16]. The incidence of DTI appears to be significantly influenced by the irradiated area and the proximity to the patient’s upper airway. In a study conducted by Deepshikha and colleagues, post-radiotherapy difficult endotracheal intubation was observed in 50% of patients [17]. In contrast to the data described in the literature on post-RT difficulties in advanced airway management for head and neck tumors, our study identified no evidence of difficult advanced airway management [18,19]. Trismus is a major threat to adequate airway management. The trismus and restricted mobility of the neck caused by RT do not improve with induction of anesthesia or muscle relaxant, unlike pain- or inflammation-induced trismus. In our study, the field of radiation covers the upper airway and, as a result, short- and long-term radiation sequelae are well known and include trismus, temporomandibular joint fibrosis, xerostomia, dysphagia, osteoradionecrosis, persistent upper airway edema, and aspiration pneumonia [16,20]. In one study, it was found that 22.3% of patients who underwent RT for head and neck carcinoma experienced restricted neck movement compared with 11.0% of patients who did not receive RT. Additionally, a higher percentage of patients in the RT group experienced trismus (24.8% versus 18.7%) [18].

In our study, we found no evidence of impaired laryngoscopy, and the mean C/L grade was lower than in other studies in the literature [17]. Mouth opening was also restricted in only 1.8% of the patients in our study. In addition, neck fibrosis and radiation changes are two of the most important predictors of impossible mask ventilation [20]. The combination of difficult or impossible mask ventilation and difficult tracheal intubation is one of the most feared complications in anesthesiology. Furthermore, the so-called “cannot intubate, cannot oxygenate situation” (CICO) is an imminent cause of anesthesia-related mortality [14]. In a recent study by Huang et al., the rate of failed tracheal intubations in cancer patients treated with RT was about ten times higher than the rates in a surgical general population [15]. Interestingly, these difficulties occurred despite the use of modern anesthesiologic techniques such as video laryngoscopy and fiberoptic endotracheal intubation techniques.

Currently, it is unclear to what extent the anatomical localization of the cancer plays a role in making airway protection more difficult. Large comparative studies on this topic are lacking in the literature. It is clinically recognized that RT targeting the larynx can cause more side effects the closer the target area is to the larynx or neck/face. A major problem in anesthesiology is that all prediction parameters or scores for DTI are not very sensitive or specific. In the above-mentioned study by Huang et al., the sensitivity and specificity of the preoperative airway assessment with regard to DTI in patients with nasopharyngeal carcinoma was also low (54.8% sensitivity and 63.9% specificity) [16].

The incidence of DTI is contingent upon a number of variables. For instance, some studies have indicated that gender is an independent predictor of difficult intubation, with women exhibiting a lower risk than men [21,22,23]. The age of the patients also seems to have an effect on the DTI. A novel finding is the identification of a relatively low risk of difficult or failed intubation in older age groups (>75 years). This is in contrast to previous work and scoring systems that included older age (defined as over 55, 57, or 60 years) as an independent predictor of difficult intubation [22,24]. In a separate study, older age was found to be associated with a greater degree of difficulty in laryngoscopy and, therefore, intubation [25].

The current literature indicates that comorbidities, including increased physical ASA status and obesity, are also strongly associated with difficult or failed intubation. This finding is in accordance with the results of previous research. In a similar vein, Ezri et al. demonstrated that an ASA health status of 3 or higher constituted an independent risk factor for difficult intubation [22]. The NAP4 report revealed that the majority of serious complications related to airway management occurred in patients with an ASA disease status of 1 or 2 (56%), compared with 46% in this cohort [26]. Furthermore, patient body weight has an impact on the ASA classification. A study by Schnittker et al. revealed an elevated risk of difficult or failed intubation in patients with obesity, a finding that aligns with the NAP4 report, which indicated that obesity is twice as prevalent in this patient population compared with the general U.K. population [19]. Nevertheless, the evidence is inconclusive regarding the role of obesity as a risk factor for difficult intubation. A number of studies have identified obesity as an independent risk factor for difficult airway management [19,23,27,28]. However, other studies have found that obesity alone is not a predictive factor for difficult intubation [29,30,31].

A review of the anesthesia literature has revealed a correlation between the number of attempts made at tracheal intubation and an increased incidence of adverse events [32,33]. A study by Sakles et al. demonstrated that successful intubation in the initial attempt was associated with a relatively low incidence of adverse events. As the number of intubation attempts increased, so did the incidence of adverse events [34].

In our findings, only 7.2% of patients required fiberoptic intubation, and 12.7% required video laryngoscopy. In addition, 96.4% of patients were successfully intubated on the first attempt. The first pass success (FPS) rates for the majority of direct laryngoscopy in our study are very high compared with data from non-RT patients and elective advanced airway management.

For example, a recent study comparing direct laryngoscopy with video laryngoscopy demonstrated an FPS rate of 82% for direct laryngoscopy in a patient collective without evidence of impaired airway management [35]. Further data indicate that the FPS rates are significantly lower, at 71%, in a mixed patient collective undergoing direct laryngoscopy for rapid sequence induction [36]. The success rates for endotracheal intubation also depend heavily on the experience of the operator. Our center has great expertise in the treatment of patients with difficult airway conditions. This could be a possible reason for the high success rates in our study. Furthermore, in most airway management studies, patients with an expected difficult airway are excluded, which can negatively influence the comparability of the studies.

One reason why the rate of difficult intubation was low in our study is that in the group of patients we examined, the radiation field was not located directly next to the larynx. In conventional RT, esophageal carcinoma was commonly included using 3–4 cm margins in the craniocaudal direction from the GTV (gross tumor volume) and following the course of the esophagus and a 1 cm margin in the lateral and anteroposterior directions from the GTV [37].

In the current study, the tumor was found to be situated at a mean distance of 21 cm from the tooth row. The height of the tumor and the associated height of the radiation may also influence the post-RT changes in the anatomical airway. Scattered radiation to the larynx depends on the irradiated volume, type of irradiated organ, and radiation technique, and this can lead to various types of damage to the larynx [38]. One potential explanation for the low-dose exposure of the larynx or trachea is the use of modern RT techniques, such as intensity-modulated radiotherapy (IMRT). This approach may result in comparatively lower DMax values than those in other published patient cohorts.

Because of the increasing success of cancer therapy and subsequent prolonged survival of cancer patients, the likelihood of these patients requiring advanced airway management for other surgeries or procedures after RT is also rising. Therefore, in the present study, we examined all patients with subsequent surgery requiring intubation, even if this was not primarily due to esophageal cancer, and with a median time of 334.7 days between RT and surgery. The time interval between RT and surgery could have an influence on the anatomical changes in the airways and should be taken into account.

Because of these complicating risks, during the initial consultation at the Department of Anesthesiology, it is important to inquire whether there has been a previous RT. Table 5 shows the airway difficulties that can occur in cancer patients that are relevant to the anesthetist.

Several limitations were acknowledged in our study, including its retrospective nature, the fact that it was conducted at a single center, and the relatively small number of patients. In the present study, there was a wide range from the time of RT to the time of surgery. This may affect the results and should be standardized in future studies. Additionally, the success of tracheal intubation is heavily reliant on the technique used and the experience of the anesthesiologists. It should be noted that our center has a high level of expertise in treating patients after RT, so the results of this study may not be generalizable to physicians with different skill levels. The results of this study demonstrate that advanced airway management was not challenging in patients undergoing surgery following radiotherapy for esophageal carcinoma in the upper and middle sections of the esophagus.

## 5. Conclusions

It is well known that radiotherapy is associated with toxic side effects on various organ systems. These effects can influence the perioperative care provided by anesthetists, particularly in advanced airway management. Our study did not identify any evidence of impaired advanced airway management in patients with esophageal cancer who had undergone radiotherapy. Nevertheless, an advanced airway management strategy is required as radiotherapy may exacerbate tumor-related airway management difficulties. The preoperative assessment should comprise a structured airway examination, including risk stratification and a physical examination. This will help to identify any toxic side effects of radiotherapy and enable effective planning of the anesthetic procedure prior to surgery to increase patient safety.

## Figures and Tables

**Figure 1 cancers-16-02540-f001:**
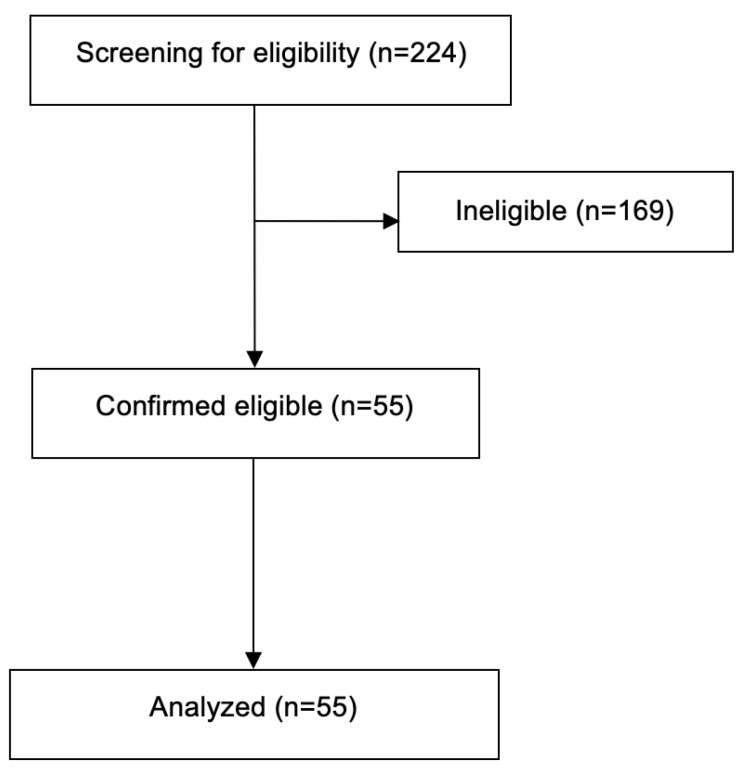
Flowchart of patient recruitment showing numbers of screened, included, and excluded patients based on the study criteria.

**Figure 2 cancers-16-02540-f002:**
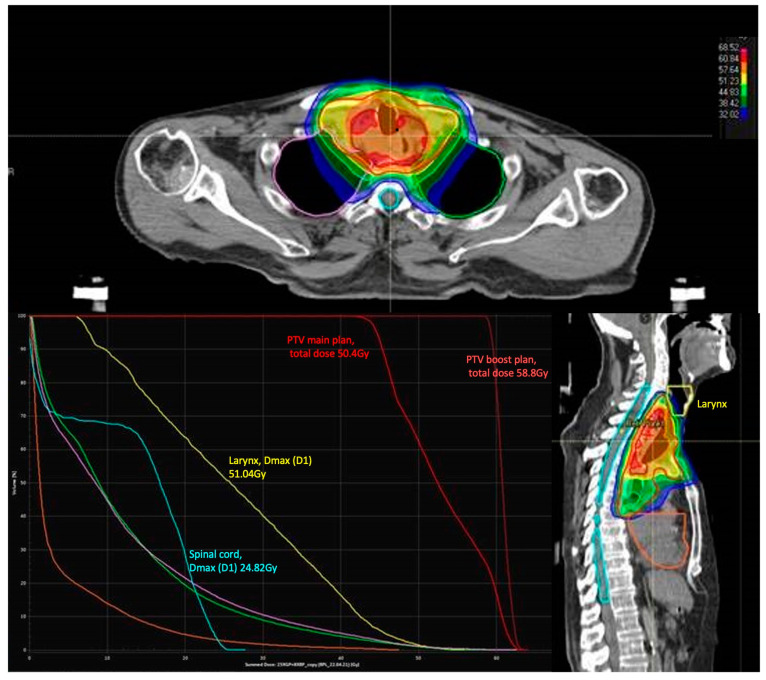
Radiation plan for a 52-year-old patient with esophageal cancer in the upper esophagus. The larynx is marked in yellow with a maximum dose (D1) of 51.04 Gy.

**Table 1 cancers-16-02540-t001:** Patient characteristics.

Characteristics	No. of Patients (%)
**Gender**	
*Female*	19 (34.5%)
*Male*	36 (65.5%)
**Body size**	
*Mean (range)*	172 cm (150–190 cm)
**Body weight**	
*Mean (range)*	70.4 kg (37.6–110 kg)
**Body Mass Index**	
*Mean (range)*	23.6 kg/m^2^ (15.3–36.3 kg/m^2^)
**Age at RT**	
*Mean (range)*	64 (49–82 years)
**Localization (cm, tooth row)**	
*Mean (range)*	21 cm (10–33 cm)
**Histology**	
Adenocarcinoma	2 (3.7%)
Squamous cell carcinoma	53 (96.3%)
**T-stage**	
*T1*	1 (1.8%)
*T2*	2 (3.6%)
*T3*	38 (69.1%)
*T4*	14 (25.5%)
**N-stage**	
*N0*	10 (18.2%)
*N+*	45 (81.8%)
**M-stage**	
*M0*	50 (90.9%)
*M1*	5 (9.1%)
**Cardiovascular risk factors**	
*Hypertension*	26 (47.2%)
*Pulmonary diseases*	9 (16.3%)
*Diabetes mellitus*	8 (14.5%)
*Smoking*	31 (56.4%)
*Coronary artery disease*	8 (14.5%)
*Heart failure*	2 (3.6%)

**Table 2 cancers-16-02540-t002:** Radiotherapy treatment characteristics.

*Radiotherapy Treatment Characteristics*	*n* (%)
**RT technique**	
*3D-CRT*	8 (14.5%)
*IMRT*	47 (85.5%)
**Mean total dose primary tumor and lymph nodes**	48.2 Gy (range:41.4–58.8 Gy)
**Mean single dose primary tumor and lymph nodes**	1.8 Gy (range:1.8–3.0 Gy)
**Mean total dose larynx**	40.0 Gy (range 14.3–68.0 Gy)
**Irradiation cervical lymph nodes**	
*Yes*	55 (100.0%)
*No*	0 (0%)
**Mean PTV volume**	844.6 ccm (range 322–1322 ccm)
**Mean time RT until surgery**	334.7 days (range 20–3787 days)
** *Reason for surgery* **	
Percutaneous gastrostomy	8 (14.5%)
Esophageal resection	26 (47.3%)
Regional lymph node resection	2 (3.6)
Others	19 (34.5)

**Table 3 cancers-16-02540-t003:** Anesthesia variables of the patients.

*Anesthesia Characteristics*	*n* (%)
**ASA physical status**	
*I (Healthy)*	0 (0.0%)
*II (Mild systemic illness)*	15 (27.2%)
*III (Severe systemic illness)*	45 (81.8%)
*IV (* *Life-threatening systemic* *illness* *)*	2 (3.6%)
**Mallampati score**	
*I (Soft palate, uvula, pillars visible)*	23 (41.8%)
*II* *(Soft palate, major part of uvula visible)*	20 (36.3%)
*III (Soft palate, base of uvula visible)*	10 (9.0%)
*IV (Only hard palate visible)*	2 (3.6%)
**Mouth opening**	
*<3* cm	1 (1.8%)
*>3* cm	54 (98.1%)
**Intubation technique**	
*Direct laryngoscopy*	44 (80.0%)
*Video laryngoscopy*	7 (12.7%)
*Fiberoptic*	4 (7.2%)
*Tracheostomy*	0 (0.0%)
**First pass intubation success**	
*Yes*	53 (96.4%)
*No*	2 (3.6%)
**Cormack/Lehane grade**	
*I*	43 (78.2%)
*II*	9 (16.4%)
*III*	3 (5.4%)
*IV*	0 (0.0%)
**Hypnotic drugs**	
*Propofol*	53 (96.3%)
*Etomidate*	2 (3.7%)
**Analgetic drugs**	
*Fentanyl*	8 (14.5%)
*Sufentanil*	41 (74.5%)
*Remifentanil*	6 (10.9%)
**Neuromuscular blocking drugs**	
*Rocuronium*	34 (61.8%)
*Mivacurium*	9 (16.4%)
*Atracurium*	2 (3.6%)
*Suxamethonium*	2 (3.6%)
*none*	8 (14.5%)

**Table 4 cancers-16-02540-t004:** Summary of complications during anesthesia.

*Complications during Anesthesia*	*n* (%)
**Oropharyngeal bleeding**	
*Yes*	1 (1.8%)
*No*	54 (98.2%)
**Severe hypoxemia**	
*Yes*	0 (0.0%)
*No*	55 (100%)
**Aspiration**	
*Yes*	0 (0.0%)
No	55 (100%)
**Peri-operative mortality**	
*Yes*	0 (0.0%)
*No*	55 (100%)

**Table 5 cancers-16-02540-t005:** Types of airway difficulties in cancer patients [14].

Mask Ventilation	Supraglottic Airway Device	Laryngoscopy
Reduced neck movement	Reduced mouth opening	Reduced mouth opening
Facial deformities	Friable oral mass occupying almost whole of the oral cavity	Deformity or growth in the mouth entrance itself
Missing teeth		Reduced subluxation of the lower jaw
		Altered neck condition
Increasing age (age > 55 years)		Distorted oral cavity

## Data Availability

The authors confirm that the data supporting the findings of this study are available within the article. Because of the nature of this research, participants of this study did not agree to share their data publicly, so supporting data are not available.

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
