# Peer review of "Impact of Radiotherapy on Endotracheal Intubation Quality Metrics in Patients with Esophageal Cancer: A Challenge for Advanced Airway Management?"

_cancers, 2024, doi:10.3390/cancers16142540_

Round 1

Reviewer 1 Report

Comments and Suggestions for Authors

Thank you for the opportunity to review this paper, I think its well written.

I jsut have a few questions:

- why not include ENT cancers with a higher dose of RT?

- or at least mention such cases and available literature thereof?

- did you observe any esophagotracheal fistulas?

Comments on the Quality of English Language

  none

Author Response

Point by Point Response to Reviewer 1:

Dear reviewer, thank you very much for your review of our manuscript. We thank you for the relevant questions and will of course respond to all of them. We were able to improve our manuscript considerably thanks to your feedback.

  1. Why not include ENT cancers with a higher dose of RT?

Following a literature search, we found that there is no data on oesophageal carcinoma after radiotherapy in the literature. From our point of view, the dose at the larnyx is particularly relevant for high-seated oesophageal tumours. We have therefore focussed on oesophageal carcinomas in this analysis. Data on ENT cancers are already available in the literature, and we agree with you that further research should also be conducted in this field.

  1. Or at least mention such cases and available literature therefo

We agree with your comment and have included and discussed the data on ENT cancers in our manuscript.

  1. Did you observe any esophagotracheal fistulas?

Thank you for the important enquiry. No oesophago-tracheal fistulas were detected in the collective examined by us, even in the follow-up examinations. We have also included this point in our manuscript.

Reviewer 2 Report

Comments and Suggestions for Authors

Dear authors,

Thank you very much for reviewing your manuscript. I am providing the following comment to address your manuscript and enhance our understanding of your research.

Major Questions:

1. What were the main goals of the research on how radiation treatment affected the quality metrics of endotracheal intubation in patients with esophageal cancer?

2. What were the main conclusions of the research on the success rate of endotracheal intubation attempts on the first try in patients with esophageal cancer treated with radiation therapy?

3. What effects did the age, gender, and ASA class of the study population have on the results of the studies on radiation and airway management in patients with esophageal cancer?

4. What were the doses of radiation given to the research participants' larynx, lymph node metastases, and main tumour?

5. Which laryngoscopy techniques were used in the study, and what was the proportion of each technique that was used?

Minor Questions:

1. What vacuum in the body of knowledge on the effects of radiation treatment on airway management and esophageal cancer does this study attempt to fill?

2. What was the duration of the study's retrospective analysis of esophageal cancer patients who received surgery and radiation treatment?

3. What proportion of research participants needed fiberoptic intubation and videolaryngoscopy in order to be successfully endotracheally intubated?

4. How does the study's conclusion on the likelihood of post-radiation airway management issues relate to the results of previous research on different forms of cancer?

5. For the study's main tumour and lymph node metastases, what were the mean prescription cumulative total dosage and mean single dose?

Best Regards

Author Response

Point by Point Response to Reviewer 2:

Dear reviewer, thank you very much for your review of our manuscript. We thank you for the relevant questions and will of course respond to all of them. We were able to improve our manuscript considerably thanks to your feedback.

 Major Questions:

  1. What were the main goals of the research on how radiation treatment affected the quality metrics of endotracheal intubation in patients with esophageal cancer?

We know from the literature and clinical practice that patients treated with radiotherapy to the neck have an increased incidence of difficult airway management. Unfortunately, there is no data in the literature on this topic in patients with oesophageal cancer. Data are available for other tumour entities, such as ENT cancers, which also show an increased difficulty of endotracheal intubation in these patients.

Therefore, our main aim was to determine whether there is evidence of more difficult airway management in patients with oesophageal cancer.

  1. What were the main conclusions of the research on the success rate of endotracheal intubation attempts on the first try in patients with esophageal cancer treated with radiation therapy?

It is now recognised in the literature that, in the best case scenario, the first intuition attempt is successful. This can significantly reduce complications during airway management. For this reason, we have investigated this variable. In our study group, there was no evidence of difficult airway management or multiple intubation attempts in patients after oesophageal radiotherapy. We can easily prove this with our data, as all airway management in our clinic is documented by the anaesthetist performing the procedure as part of the medical documentation required in Germany. Our in-house standard is to document the details of intubation, such as Cormack/Lehane score, or injuries and number of attempts.

  1. What effects did the age, gender, and ASA class of the study population have on the results of the studies on radiation and airway management in patients with esophageal cancer?

A review of the literature reveals that the incidence of difficult airway management is dependent on a number of factors. Such factors include age, gender and ASA classification. This point has been included in the discussion. As this aspect is dependent on numerous factors already discussed, it is not possible to make any definitive statements regarding it in the context of our sample. Nevertheless, the collation of baseline data from patients is of paramount importance in order to facilitate comparison between studies. Further studies should be conducted in order to identify causal relationships.

  1. What were the doses of radiation given to the research participants' larynx, lymph node metastases, and main tumour?

Thank you for the important question. We have adjusted and clarified our table.

The prescribed cumulative RT-dose to primary tumor and lymph node metastasis was 48.2Gy (range:41.4Gy-58.8Gy). The median single dose to primary tumor and lymph node metastasis were 1.8Gy (range:1.8Gy-3.0Gy). The mean total dose to larynx was 40.0Gy (range 14.3Gy-68.0Gy).

  1. Which laryngoscopy techniques were used in the study, and what was the proportion of each technique that was used?

Thank you for the important question. We have shown this in table 3. In our study, direct/conventional larnygoscopy was used for intubation, as well as video larnygoscopy and fibreoptic intubation. Details and frequencies can be found in Table 3.

Minor Questions:

  1. What vacuum in the body of knowledge on the effects of radiation treatment on airway management and esophageal cancer does this study attempt to fill?

The objective of this publication is to enhance the scientific profile of this tumour entity and to facilitate its inclusion in future studies. A greater number of cases and multi-centre studies could facilitate a more definitive statement on this topic. Additionally, the objective of the study was to enhance the awareness of physicians engaged in the treatment of oesophageal cancer and to identify patients who have undergone radiotherapy of the oesophagus as high-risk individuals. However, our findings indicate that there is no evidence to support this hypothesis in a limited number of cases.

  1. What was the duration of the study's retrospective analysis of esophageal cancer patients who received surgery and radiation treatment?

Patients who were irradiated and underwent surgery with esophageal cancer in Heidelberg,Germany from 2012 to 2023 were included. In the present study there was a mean follow-up of 49 months.

  1. What proportion of research participants needed fiberoptic intubation and videolaryngoscopy in order to be successfully endotracheally intubated?

Thank you for the important question. We have shown the number of different methods used in our table 3.

  1. How does the study's conclusion on the likelihood of post-radiation airway management issues relate to the results of previous research on different forms of cancer?

We did not show an increased incidence of difficult airway management compared to other head and neck cancers. We have included this aspect in our discussion and explained it in more detail.

  1. For the study's main tumour and lymph node metastases, what were the mean prescription cumulative total dosage and mean single dose?

Thank you for the important question. The prescribed cumulative RT-dose to primary tumor and lymph node metastasis was 48.2G (range:41.4Gy-58.8Gy). The median single dose to primary tumor and lymph node metastasis were 1.8Gy (range:1.8Gy-3.0Gy). We include this sentence in Result and Material and methods part as well as in table 2.

Round 2

Reviewer 1 Report

Comments and Suggestions for Authors

thank you for your revisions

Reviewer 2 Report

Comments and Suggestions for Authors

Dear auther,

Thank you very much for addressing all the comments in your manuscript.

Best Regards